# Generalized Reduction to the Isotropy for Flexible Equivariant Neural Fields

**Alejandro García-Castellanos**[1]* **Gijs Bellaard**[2]
**Remco Duits**[2] **Daniël M. Pelt**[3] **Erik J. Bekkers**[1]
[1]Amsterdam Machine Learning Lab (AMLab), University of Amsterdam
[2]Department of Mathematics and Computer Science, Eindhoven University of Technology
[3]Leiden Institute of Advanced Computer Science, Universiteit Leiden

## Abstract

Many geometric learning problems require invariants on heterogeneous product spaces, i.e., products of distinct spaces carrying different group actions, where standard techniques do not directly apply. We show that, when a group $G$ acts transitively on a space $M$, any $G$-invariant function on a product space $X \times M$ can be reduced to an invariant of the isotropy subgroup $H$ of $M$ acting on $X$ alone. Our approach establishes an explicit orbit equivalence $(X \times M)/G \cong X/H$, yielding a principled reduction that preserves expressivity. We apply this characterization to Equivariant Neural Fields, extending them to arbitrary group actions and homogeneous conditioning spaces, and thereby removing the major structural constraints imposed by existing methods.

## 1 Introduction

Symmetry is a powerful structural prior in geometric machine learning. When data admit a known group of transformations, building invariance or equivariance directly into model architectures improves generalization and data efficiency (Bronstein et al., 2021; Vadgama et al., 2025). A fundamental object in this context is the *joint invariant*: a function unchanged under the simultaneous action of a group on multiple inputs.

Formally, let $G$ be a group acting on spaces $X_1, \ldots, X_m$. The diagonal action on the product $X_1 \times \cdots \times X_m$ is given by $g \cdot (x_1, \ldots, x_m) := (g \cdot x_1, \ldots, g \cdot x_m)$, and a joint invariant is any function $f_G : X_1 \times \cdots \times X_m \to \mathbb{R}$ satisfying $f_G(g \cdot x_1, \ldots, g \cdot x_m) = f_G(x_1, \ldots, x_m)$ for all $g \in G$. Such invariants serve both as building blocks for invariant models and as key ingredients in equivariant map constructions (Villar et al., 2021; Kaba et al., 2023; Bekkers et al., 2023).

Most existing work focuses on products $X^{\times m}$, where all factors are copies of a single space $X$. This setting arises naturally in point cloud processing, where inputs consist of $m$ points in $\mathbb{R}^d$ (Blum-Smith et al., 2024; Dym & Gortler, 2023; Villar et al., 2021), and complete characterizations of such joint invariants are known for many $(G, X)$ pairs. However, many learning problems involve *heterogeneous product spaces*, i.e., products of distinct spaces carrying different group actions. Here, invariant constructions are less straightforward and typically require ad hoc, problem-specific design (Knigge et al., 2024). Additional background appears in the Appendix A.

A motivating example arises in *Equivariant Neural Fields (ENFs)* (Wessels et al., 2024), which represent signal families via coordinate-based networks $f_\theta : X \times \mathcal{Z} \to \mathbb{R}^d$, where $X$ denotes spatial coordinates and $\mathcal{Z}$ is a latent conditioning space. Equivariance is imposed through the constraint $f_\theta(g \cdot x, g \cdot z) = f_\theta(x, z)$, reducing the problem to constructing joint invariants on the heterogeneous product $X \times \mathcal{Z}$. Current ENF architectures handle only specific groups and spaces, leaving the general case open (García-Castellanos et al., 2025).

In this work, we develop a systematic framework for joint invariants on heterogeneous products. Our main tool is the *Generalized Reduction to the Isotropy*, which exploits the structure present when one factor is a homogeneous space, i.e., when the group acts transitively on it. Under this

---

*Corresponding author: `<a.garciacastellanos@uva.nl>`

assumption, invariants on the full product reduce to invariants of a smaller isotropy subgroup acting on the remaining factors, yielding a principled approach to invariant design.

We demonstrate the utility of this framework on Equivariant Neural Fields, where it yields maximally expressive invariants and enables more flexible architectures. Concretely, our results extend the ENF's framework of García-Castellanos et al. (2025)—which is currently the most general approach for ENFs with provable expressiveness guarantees—to arbitrary group actions on $X \times \mathcal{Z}$, for arbitrary input spaces $X$ and arbitrary homogeneous conditioning spaces $\mathcal{Z}$. More broadly, our reduction to the isotropy framework is applicable beyond ENFs, and we discuss further applications in machine learning in Appendix E.

## 2 MAIN RESULT

In this section, we present the main theoretical contribution of this work: a general reduction principle that showcases how $G$-invariants on product spaces can be simplified without loss of information.

An introduction to the necessary mathematical foundations of this work can be found in Appendix B.

**Problem Formulation.** Let $G$ be a group acting transitively on a space $M$ and (not necessarily transitively) on a space $X$. We consider the diagonal action of $G$ on the product space $X \times M$ and study its invariant functions.

Our objective is to characterize all functions $f_G : X \times M \to Y$ that are invariant under this action and to construct sets of invariants that separate orbits, thereby providing a complete description of the orbit space $(X \times M)/G$ (Dym & Gortler, 2023; Olver, 1995).

**Approach Overview.** In this work, we show how this problem can be reformulated by exploiting the transitivity of the $G$-action on $M$. Fixing a reference point $p_0 \in M$ determines an isotropy subgroup $H := \mathrm{Stab}_G(p_0)$ and yields a canonical normalization of the $M$–component of each $G$-orbit in $X \times M$. Consequently, the orbit structure of the diagonal $G$-action is entirely determined by the induced action of $H$ on $X$.

The remainder of this section formalizes this observation by establishing an explicit equivalence between the orbit spaces $(X \times M)/G$ and $X/H$, and by deriving a corresponding reduction principle for invariant functions, which we refer to as the *Generalized Reduction to the Isotropy*.

### 2.1 ORBIT EQUIVALENCE

We begin by fixing the notation that will be used throughout this subsection. Let $G$ act on a set $M$. For $p, q \in M$, define $G_{p,q} := \{g \in G \mid g \cdot p = q\}$. In particular, $G_{p,p} = \mathrm{Stab}_G(p)$ is the stabilizer of $p$. For any subgroup $H \subseteq G$ and $x \in X$, we write $H \cdot x := \{h \cdot x \mid h \in H\}$ for the corresponding orbit.

The following lemma establishes a fundamental bijection between orbit spaces that underlies our main results.

**Lemma 2.1** (Orbit Equivalence). *Let $G$ act transitively on the set $M$ and (not necessarily transitively) on the set $X$. Fix $p_0 \in M$ and let $H := \mathrm{Stab}_G(p_0)$. Then the map $\Phi : (X \times M)/G \to X/H$, defined by*

$$\Phi\big(G \cdot (x,p)\big) = G_{p,p_0} \cdot x \in X/H, \tag{1}$$

*is a bijection between the orbit spaces $(X \times M)/G$ and $X/H$.*

*Proof.* See Appendix C.1.

**Remark 2.1** (Homogeneous space identification). *If $M$ is a homogeneous space, i.e., it has a transitive $G$-action, then it can be identified with the coset space $G/H$ via the correspondence $p \leftrightarrow gH$, where $g \in G$ satisfies $g \cdot p_0 = p$. Under this identification, $g^{-1} \in G_{p,p_0}$, and the bijection $\Phi$ from Equation (1) takes the form*

$$\Phi\big(G \cdot (x,gH)\big) = H \cdot (g^{-1} \cdot x).$$

> **Intuition of orbit equivalence property**
>
> To build intuition for Lemma 2.1, consider rewriting the orbit $G \cdot (x, p)$ for any $p \in M$ and $x \in X$:
>
> $$G \cdot (x, p) = \{g \cdot (x, p) \mid g \in G\}$$
> $$= \bigcup_{g' \in G_{p, p_0}} \{g \cdot (g' \cdot x, p_0) \mid g \in G\}$$
> $$= \bigcup_{g' \in G_{p, p_0}} G \cdot (g' \cdot x, p_0) \tag{2}$$
> $$= \bigcup_{h \in H} G \cdot (hg' \cdot x, p_0), \quad \text{for any fixed } g' \in G_{p, p_0}. \tag{3}$$
>
> The bijection $\Phi$ associates the orbit $G \cdot (x, p)$ with the collection of second components in (2):
>
> $$G \cdot (x, p) \longleftrightarrow G_{p, p_0} \cdot x = \{g' \cdot x \mid g' \in G_{p, p_0}\}.$$
>
> Using (8) and (3), this is equivalently the $H$-orbit $H \cdot (g' \cdot x)$ for any choice of $g' \in G_{p, p_0}$.

## 2.2 GENERALIZED REDUCTION TO THE ISOTROPY

Having established the bijection between orbit spaces, we now present our main technique for relating invariants of the $G$-action on $X \times M$ to invariants of the $H$-action on $X$. In practice, computing $H$-invariants on $X$ is often substantially simpler than computing $G$-invariants on $X \times M$.

We first recall a classic result from algebra.

**Theorem 2.1** (Universal Property of Quotients; see Dummit & Foote (2003)). *Let $R$ be an equivalence relation on a set $X$, let $f_R : X \to Y$ be a function, and let $\pi : X \to \tilde{X}$ denote the quotient map $x \mapsto [x]$. Then $f_R$ is $R$-invariant if and only if there exists a unique function $\tilde{f}_R : \tilde{X} \to Y$ such that $f = \tilde{f} \circ \pi$.*

Let $\rho : M \to G$ be any map satisfying

$$\rho(p) \cdot p = p_0 \quad \text{for all } p \in M, \tag{4}$$

i.e., $\rho(p) \in G_{p, p_0}$ for every $p$. We call $\rho$ a *canonicalization map*. Note that $\rho$ need not be a moving frame in the sense of Olver (2001), since the action of $G$ on $M$ may fail to be free.

**Theorem 2.2** (Generalized Reduction to the Isotropy). *Let $\rho : M \to G$ satisfy (4), and define $T : X \times M \to X$ by $T(x, p) = \rho(p) \cdot x$. Then for every $H$-invariant function $f_H : X \to Y$, the composition*

$$f_G(x, p) := f_H \circ T(x, p) = f_H(\rho(p) \cdot x) \tag{5}$$

*defines a $G$-invariant function $f_G : X \times M \to Y$. Conversely, every $G$-invariant on $X \times M$ arises uniquely in this manner. Moreover, the resulting invariants are independent of the choice of $\rho$.*

*In particular, the following diagram commutes:*

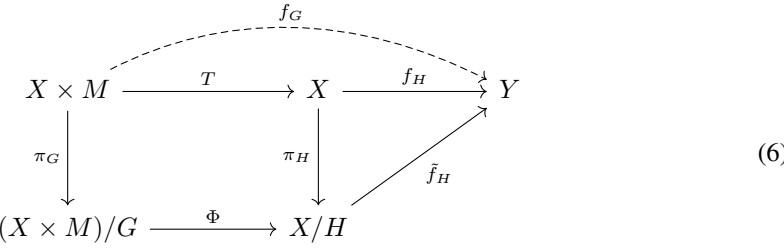

$$\tag{6}$$

*Proof.* The proof follows the same steps as Corollary 3.1 and Lemma 3.1 in Hayes (2022), replacing Theorem 3.1 therein with Lemma 2.1. $\square$

As noted in Remark 2.1, when the homogeneous space $M$ is seen as the quotient space $G/H$, a natural choice is $\rho(gH) = g^{-1}$ for any representative $g$ of the $gH$ coset.

**Remark 2.2.** *Related reduction results appear in Hayes (2022) and Bekkers et al. (2023); however, our method strictly generalizes these constructions. Specifically, Bekkers et al. (2023) characterizes invariants on $M \times M$ for a homogeneous space $M$, while Hayes (2022) extends this to $m$-fold products $M^{\times m}$. Our framework unifies both approaches within a single setting applicable to arbitrary heterogeneous products $X \times M$.*

## 2.3 PROPERTIES OF THE REDUCTION

We now establish several properties of the reduction to the isotropy that broaden its applicability; proofs are deferred to Appendix C.2. For further results, see Appendix C.2.3.

### 2.3.1 FLEXIBILITY IN CHOICE OF REDUCTION

Consider a product space $X \times M_1 \times M_2$ where $G$ acts transitively on both $M_1$ and $M_2$. Fix points $p_0 \in M_1$ and $q_0 \in M_2$, and let $H_1 = \mathrm{Stab}_G(p_0)$ and $H_2 = \mathrm{Stab}_G(q_0)$. One may reduce along either homogeneous space: invariants of $H_1$ acting on $X \times M_2$ correspond to invariants of $H_2$ acting on $X \times M_1$. This flexibility allows one to choose the reduction that yields the simplest computation.

**Corollary 2.1.** *Let $\Phi_1 : (X \times M_1 \times M_2)/G \to (X \times M_2)/H_1$ and $\Phi_2 : (X \times M_1 \times M_2)/G \to (X \times M_1)/H_2$ be defined as in (1), with corresponding maps $T_1$ and $T_2$. Then for any $H_2$-invariant function $f_{H_2} : X \times M_1 \to Y$, the identity*

$$f_{H_1} = \tilde{f}_{H_2} \circ \Phi_2 \circ \Phi_1^{-1} \circ \pi_{H_1}$$

*defines a unique $H_1$-invariant function $f_{H_1} : X \times M_2 \to Y$ satisfying*

$$f_{H_1} \circ T_1 = f_{H_2} \circ T_2.$$

### 2.3.2 THE GROUP AS A HOMOGENEOUS SPACE

When $M = G$ itself (viewed as a homogeneous space under left multiplication), the stabilizer of the identity is trivial, and we obtain a complete characterization of $G$-invariants on $X \times G$

**Corollary 2.2.** *Let $X \times G$ be equipped with the diagonal $G$-action $(g, (x, h)) \mapsto (g \cdot x, gh)$. Then every $G$-invariant function $f_G : X \times G \to Y$ is of the form*

$$f_G(x, g) = f(g^{-1} \cdot x)$$

*for some function $f : X \to Y$.*

**Remark 2.3.** *Corollary 2.2 generalizes Theorem 4.1 of García-Castellanos et al. (2025), which assumes $G$ is a Lie group and $X$ is a product of Riemannian manifolds with regular $G$-actions. Our formulation imposes no such restrictions. Additionally, it provides a formal justification for the energy-based canonicalization formulation introduced in Equation (6) of Kaba et al. (2023).*

## 3 USE CASE: EQUIVARIANT NEURAL FIELDS

We demonstrate the practical utility of our framework by extending the Equivariant Neural Eikonal Solver of García-Castellanos et al. (2025), an instance of Equivariant Neural Fields (ENFs) (Wessels et al., 2024). On a Riemannian manifold $\mathcal{X}$ endowed with a velocity field, the Eikonal equation defines a travel-time function $T : \mathcal{X} \times \mathcal{X} \to \mathbb{R}^+$ that encodes shortest arrival times between source–receiver pairs. In this setting, the Equivariant Neural Field $f_\theta : \mathcal{X} \times \mathcal{X} \times \mathcal{Z} \to \mathbb{R}^+$ represents a family of Eikonal solutions and is conditioned on instance-specific latent variables $z \in \mathcal{Z}$. Each latent variable corresponds to a particular travel-time solution induced by a fixed velocity field.

As discussed in Section 1, the central challenge in designing such ENFs lies in constructing $G$-invariant functions on the heterogeneous product $\mathcal{X} \times \mathcal{X} \times \mathcal{Z}$. Prior work (García-Castellanos et al., 2025) sidesteps this challenge by restricting the latent space to the group itself ($\mathcal{Z} = G$). Theorem 2.2 removes this restriction, enabling conditioning on arbitrary homogeneous spaces of the form $\mathcal{Z} = G/H$.

---

**Algorithm 1:** Computing Separating $G$-Invariants on $X^{\times m} \times G/H$

---

**Input:** Group $G$, subgroup $H \leq G$, $G$-space $X$, $m \in \mathbb{N}$
**Output:** Separating $G$-invariants on $X^{\times m} \times G/H$

$\triangleright$ `Reduce to` $H$`-invariants on` $X^{\times m}$ `(Thm. 2.2)`
Let $\{f_H^\ell(x_1, \ldots, x_m)\}_{\ell=1}^L$ be separating $H$-invariants      `// e.g. via moving frame`

$\triangleright$ `Lift to` $G$`-invariants via canonicalization`
$\rho\colon G/H \to G, \quad \rho(gH) := g^{-1}$      `// for any representative g of gH`

**return** $\left\{ (x_1, \ldots, x_m, gH) \mapsto f_H^\ell\big(g^{-1} \cdot x_1, \ldots, g^{-1} \cdot x_m\big) \right\}_{\ell=1}^L$

---

Crucially, our *generalized reduction to the isotropy* transforms the original problem into that of constructing $H$-invariant functions on $\mathcal{X}^{\times m}$, a classical setting that admits powerful tools from invariant theory, including the moving frame method and Weyl's theorem (see Appendix A).

Algorithm 1 operationalizes this reduction into a practical procedure for constructing separating $G$-invariants, which ensure maximal expressivity for invariant function approximation (see Proposition B.1). Explicit instantiations for 2D and 3D Euclidean and spherical geometries, under various choices of latent space, are provided in Appendix D.

## 4   DISCUSSION AND FUTURE WORK

We presented a *generalized reduction to the isotropy* principle: if $G$ acts transitively on $M$, then there is a canonical bijection of orbit spaces $(X \times M)/G \cong X/H$, where $H$ is an isotropy subgroup. This identification relates $G$-invariants on heterogeneous products to $H$-invariants on a reduced space, and shows that these computations can be transferred without information loss.

We demonstrated the practical utility of this framework by extending Equivariant Neural Fields to arbitrary homogeneous conditioning spaces, removing a significant structural limitation of prior work. More broadly, our reduction principle applies to other domains where symmetries are a powerful inductive bias, such as equivariant reinforcement learning (see Appendix E for further discussion).

A systematic empirical evaluation comparing different choices of conditioning space and their effects on learning dynamics remains an important direction for future work.

## ACKNOWLEDGEMENTS

Alejandro García Castellanos is funded by the Hybrid Intelligence Center, a 10-year programme funded through the research programme Gravitation which is (partly) financed by the Dutch Research Council (NWO). This publication is part of the project SIGN with file number VI.Vidi.233.220 of the research programme Vidi which is (partly) financed by the Dutch Research Council (NWO) under the grant `https://doi.org/10.61686/PKQGZ71565`. Remco Duits and Gijs Bellaard gratefully acknowledge NWO for financial support via VIC.202.031.

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

## A    RELATED WORK

As discussed in Section 1, the most commonly studied setting in machine learning for joint invariants involves products $X^{\times m}$, where all factors are copies of a single space. In this setting, numerous techniques have been developed for computing invariants, including Weyl's theorem (Weyl, 1946; Villar et al., 2021), infinitesimal methods (Andreassen & Kruglikov, 2020), moving frames (Olver, 2001), differential invariants (Olver, 1995; Sangalli et al., 2022), representer-based approaches (Bellaard et al., 2025), modern canonicalization (Kaba et al., 2023; Shumaylov et al., 2024), and frame averaging methods (Puny et al., 2022). Additionally, sketching techniques have been proposed to reduce the number of required invariants while preserving maximal expressivity (Dym & Gortler, 2023).

While some of these techniques can be adapted to heterogeneous products, several powerful methods, such as Weyl's theorem, are not directly applicable to this more general setting. This is a significant limitation, as Weyl's theorem has been a key component in many modern applications of invariant theory to machine learning (Dym & Gortler, 2023; Villar et al., 2021). However, as we demonstrate in Appendix D, our proposed *Generalized Reduction to the Isotropy* restores the applicability of these techniques in many common scenarios, enabling practitioners to employ their preferred methodology.

## B    MATHEMATICAL PRELIMINARIES

In this section, we present the necessary mathematical foundations. For a comprehensive treatment of these topics, we refer the reader to Dummit & Foote (2003) and Olver (1995).

We are interested in (symmetry) groups, which are algebraic constructs that consist of a set $G$ and a group product—which we denote as a juxtaposition—that satisfies certain axioms, such as the existence of an identity element $e \in G$ such that for all $g \in G$ we have $eg = ge = g$, closure such that for all $g, h \in G$ we have $gh \in G$, the existence of an inverse $g^{-1}$ for each $g$ such that $g^{-1}g = e$, and associativity such that for all $g, h, i \in G$ we have $(gh)i = g(hi)$. A Lie group $G$ is a smooth manifold with group operations that are smooth.

A (left) group action on a set $X$ is a map $\mu : G \times X \to X$ satisfying $\mu(e, x) = x$ and $\mu(g, \mu(h, x)) = \mu(gh, x)$ for all $x \in X$, $g, h \in G$. When $\mu$ is clear from context we write $g \cdot x$. The orbit of a point $x \in X$ under $G$ is the set $G \cdot x := \{g \cdot x \mid g \in G\}$. The orbit space $X/G$ is the quotient space obtained by identifying points in $X$ that lie in the same orbit under the $G$-action. Formally, $X/G := \{G \cdot x \mid x \in X\}$ consists of all distinct orbits, where each orbit represents an equivalence class under the relation $x \sim y \Leftrightarrow \exists g \in G$ such that $y = g \cdot x$. These equivalence classes partition $X$ into mutually disjoint subsets whose union equals $X$, yielding a canonical decomposition that reflects the underlying symmetry of the group action.

The isotropy subgroup (or stabilizer) of $G$ at $x \in X$ is $\mathrm{Stab}_G(x) := \{g \in G \mid g \cdot x = x\}$. A group $G$ acts *freely* on $X$ if $\mathrm{Stab}_G(x) = \{e\}$ for all $x \in X$, meaning no non-identity element fixes any point. A group acts *regularly* on a manifold $\mathcal{M}$ if each point has arbitrarily small neighborhoods whose intersections with each orbit are connected. In practical applications, the groups of interest typically act regularly.

A group $G$ acts *transitively* on $X$ if for any $x, y \in X$, there exists $g \in G$ such that $g \cdot x = y$. Equivalently, the action is transitive if there is exactly one orbit, i.e., $G \cdot x = X$ for any $x \in X$. A set $X$ equipped with a transitive $G$-action is called a *homogeneous space*. By the orbit-stabilizer theorem, any homogeneous space $X$ is $G$-equivariantly isomorphic to the quotient space $G/\mathrm{Stab}_G(x_0)$ for any choice of base point $x_0 \in X$; different choices of base point yield conjugate stabilizers and hence isomorphic quotients (Dummit & Foote, 2003).

For the applications in Appendix D, we consider the following matrix Lie groups acting on $\mathbb{R}^n$. The orthogonal group $\mathrm{O}(n) := \{R \in \mathbb{R}^{n \times n} \mid RR^\top = I_n\}$ acts by $R \cdot x = Rx$, and its subgroup $\mathrm{SO}(n) := \{R \in \mathrm{O}(n) \mid \det(R) = 1\}$ comprises rotations. The Euclidean group $\mathrm{E}(n) := \mathbb{R}^n \rtimes \mathrm{O}(n)$ consists of elements $g = (t, R)$ with action $g \cdot x = Rx + t$ and composition $(t, R)(t', R') = (t + Rt', RR')$; restricting to $\mathrm{SO}(n)$ yields the special Euclidean group $\mathrm{SE}(n)$ of rigid motions. Both $\mathrm{O}(n)$ and $\mathrm{SO}(n)$ act on the unit sphere $S^{n-1} \subset \mathbb{R}^n$ via the inherited action $R \cdot x = Rx$.

Moreover, we will use the following properties of invariant functions, which establish the connection between orbit separation and universal approximation.

**Definition B.1** (Orbit Separation; see (Dym & Gortler, 2023)). *Let $G$ be a group acting on a set $X$, let $Y$ be a set, and let $f : X \to Y$ be an invariant function. We say that $f$ separates orbits if $f(x) = f(y)$ implies that $x = g \cdot y$ for some $g \in G$. We say that a finite collection of invariant functions $f_i : \mathcal{X} \to \mathcal{Y}$, $i = 1, \ldots, N$ separates orbits, if the concatenation $(f_1(x), \ldots, f_N(x))$ separates orbits.*

**Proposition B.1** (Maximal Expressivity; see (Dym & Gortler, 2023)). *Let $\mathcal{X}$ be a topological space, and $G$ a group which acts on $\mathcal{X}$. Let $K \subset \mathcal{X}$ be a compact set, and let $f^{\mathrm{inv}} : \mathcal{X} \to \mathbb{R}^N$ be a continuous $G$-invariant map that separates orbits. Then for every continuous invariant function $f : \mathcal{X} \to \mathbb{R}$, there exists some continuous $f^{\mathrm{general}} : \mathbb{R}^N \to \mathbb{R}$ such that*

$$f(x) = f^{\mathrm{general}}(f^{\mathrm{inv}}(x)), \quad \forall x \in K.$$

## C    REDUCTION TO THE ISOTROPY

In this section, we present the complete proofs and additional properties for the results presented in Section 2.

### C.1    PROOF OF ORBIT EQUIVALENCE

Let a group $G$ acting on a set $M$. We define $G_{p,q} := \{g \in G \mid g \cdot p = q\}$ to be the set of all group elements mapping $p \in M$ to $q \in M$.

**Lemma 2.1 (Restated).** *Let $G$ be a group acting on transitively on the set $M$ and (not necessarily transitively) on the set $X$. Let $p_0$ be any point in $M$ and define the subgroup $H = \mathrm{Stab}_G(p_0)$ (also called an isotropy subgroup). Then, the map $\Phi : (X \times M)/G \to X/H$, given by*

$$\Phi(G \cdot (x,p)) = G_{p,p_0} \cdot x \in X/H, \tag{7}$$

*is a bijection of the orbit spaces $(X \times M)/G$ and $X/H$.*

*Proof.* We will show the bijection between $(X \times M)/G$ and $X/H$ by constructing two maps, the *forward* map $\Phi : (X \times M)/G \to X/H$ and the *backward* map $\Psi : X/H \to (X \times M)/G$, which are each others inverse.

**1) Forward map def:** Let $o \in (X \times M)/G$. Pick any $(x,p) \in o$. We define the forward map by

$$\Phi(o) = G_{p,p_0} \cdot x.$$

Notice that $G_{p,q}$ is *never* empty since $G$ acts transitively on $M$. Let's confirm that this map is well defined, meaning that the choice of $(x,p) \in o$ does *not* matter. Let's choose another candidate $(x',p')$ from $o$, then there exists $g' \in G$ such that $g' \cdot (x,p) = (x',p')$. Therefore, we would want to check whether $G_{p,p_0} \cdot x = G_{p',p_0} \cdot x'$. Let $a \in G_{p,p_0} \cdot x$, and $b \in G_{p',p_0} \cdot x'$, then

$$\begin{cases} a = g \cdot x & \text{s.t. } g \cdot p = p_0 \\ b = h \cdot x' & \text{s.t. } h \cdot p' = p_0 \end{cases}$$

$(\supset)$ Since $p_0 = h \cdot p' = hg' \cdot p$, then $hg' \in G_{p,p_0}$, and $b = h \cdot x' = hg' \cdot x \in G_{p,p_0} \cdot x$. Thus $G_{p,p_0} \cdot x \supset G_{p',p_0} \cdot x'$.

$(\subset)$ Since $p_0 = g \cdot p = g(g')^{-1} \cdot p'$, then $g(g')^{-1} \in G_{p',p_0}$, and $a = g \cdot x = g(g')^{-1} \cdot x' \in G_{p',p_0} \cdot x'$. Thus $G_{p,p_0} \cdot x \subset G_{p',p_0} \cdot x'$.

Lastly, we need to check that $\Phi(o) = G_{p,p_0} \cdot x \in X/H$. To do so, we will see that

$$G_{p,p_0} \cdot x = H \cdot (g \cdot x) \in X/H \quad \text{for any } g \in G_{p,p_0}. \tag{8}$$

Let $a \in G_{p,p_0} \cdot x$, and $b \in H \cdot (g \cdot x)$, then

$$\begin{cases} a = g' \cdot x & \text{s.t. } g' \cdot p = p_0 \\ b = hg \cdot x & \text{s.t. } g \cdot p = p_0, \text{ and } h \cdot p_0 = p_0 \end{cases}$$

$(\supset)$ Since $hg \cdot p = h \cdot p_0 = p_0$, then $hg \in G_{p,p_0}$ for all $h \in H$. Thus, $Hg \subset G_{p,p_0}$ for all $g \in G_{p,p_0}$, and therefore, $H \cdot (g \cdot x) \subset G_{p,p_0} \cdot x$ for all $g \in G_{p,p_0}$.

$(\subset)$ We want to check whether $G_{p,p_0} \subset Hg$, i.e., if given $g, g' \in G_{p,p_0}$ there exist $h \in H$ such as $g' = hg$. Notice that this is equivalent to verifying if $g'g^{-1} = h \in H$. Since $g'g^{-1} \cdot p_0 = g' \cdot p = p_0$, then we see that $g'g^{-1} \in H = \mathrm{Stab}_G(p_0)$. Thus, $G_{p,p_0} \subset Hg$ for all $g \in G_{p,p_0}$, and therefore, $G_{p,p_0} \cdot x \subset H \cdot (g \cdot x)$ for all $g \in G_{p,p_0}$.

**2) Backward map def:** Let $q \in X/H$. Pick any $x \in q$. We define the backward map by

$$\Psi(q) = G \cdot (x, p_0). \tag{9}$$

Again, let's confirm that this map is well defined, meaning that the choice of $x \in q$ does *not* matter. Let's choose another candidate $y$ from $q$, then there exists $h \in H$ such that $h \cdot x = y$. Therefore, we would want to check whether $G \cdot (x, p_0) = G \cdot (y, p_0)$. Let $a \in G \cdot (x, p_0)$, and $b \in G \cdot (y, p_0)$, then

$$\begin{cases} a = g \cdot (x, p_0) & \text{for some } g \in G \\ b = g' \cdot (y, p_0) & \text{for some } g' \in G \end{cases}$$

$(\supset)$ Since $b = g' \cdot (y, p_0) = g' \cdot (h \cdot x, h \cdot p_0) = g'h \cdot (x, p_0)$, and $H$ is a subgroup of $G$, then $b \in G \cdot (x, p_0)$. Thus, $G \cdot (x, p_0) \supset G \cdot (y, p_0)$.

$(\subset)$ Since $a = g \cdot (x, p_0) = g \cdot (h^{-1} \cdot y, h^{-1} \cdot p_0) = gh^{-1} \cdot (y, p_0)$, and $H$ is a subgroup of $G$, then $a \in G \cdot (y, p_0)$. Thus, $G \cdot (x, p_0) \subset G \cdot (y, p_0)$.

**3) Inverse verification:** We check that the forward and backward map are each other inverse. Let $o \in (X \times M)/G$. Pick any $(x, p) \in o$ and $g \in G_{p,p_0}$. We have

$$(\Psi \circ \Phi)(o) = \Psi(G_{p,p_0} \cdot x) = \Psi(H \cdot (g \cdot x)) = G \cdot (g \cdot x, p_0) = G \cdot (g \cdot (x, p)) = G \cdot (x, p) = o.$$

Now let $q \in X/H$. Pick any $x \in q$. Notice that $G_{p,p} = \mathrm{Stab}_G(p) = \{g \mid g \cdot p = p\}$. Thus, we have

$$(\Phi \circ \Psi)(q) = \Phi(G \cdot (x, p_0)) = G_{p_0,p_0} \cdot x = H \cdot x = q.$$

So, indeed, the forward and backward map are each other inverses, showing that $(X \times M)/G$ and $X/H$ are in bijection.

$\square$

### C.2 PROPERTIES OF REDUCTION TO THE ISOTROPY

#### C.2.1 PROOF OF FLEXIBILITY OF CHOICE OF REDUCTION

**Corollary 2.1 (Restated).** *Let $\Phi_1 : (X \times M_1 \times M_2)/G \to (X \times M_2)/H_1$, and $\Phi_2 : (X \times M_1 \times M_2)/G \to (X \times M_1)/H_2$ defined as in Equation (1), and let $\rho_1 : M_1 \to G$, and $\rho_2 : M_2 \to G$ be canonicalization maps satisfying the identity in Equation (4), and their corresponding induced maps $T_1 : X \times M_1 \times M_2 \to X \times M_2$, and $T_2 : X \times M_1 \times M_2 \to X \times M_1$.*

*Then for a given $H_2$ invariant $f_{H_2} : X \times M_1 \to Y$, the identity*

$$f_{H_1} = \tilde{f}_{H_2} \circ \Phi_2 \circ \Phi_1^{-1} \circ \pi_{H_1}$$

*defines a unique $H_1$ invariant function $f_{H_1} : X \times M_2 \to Y$, such that the following diagram commutes*

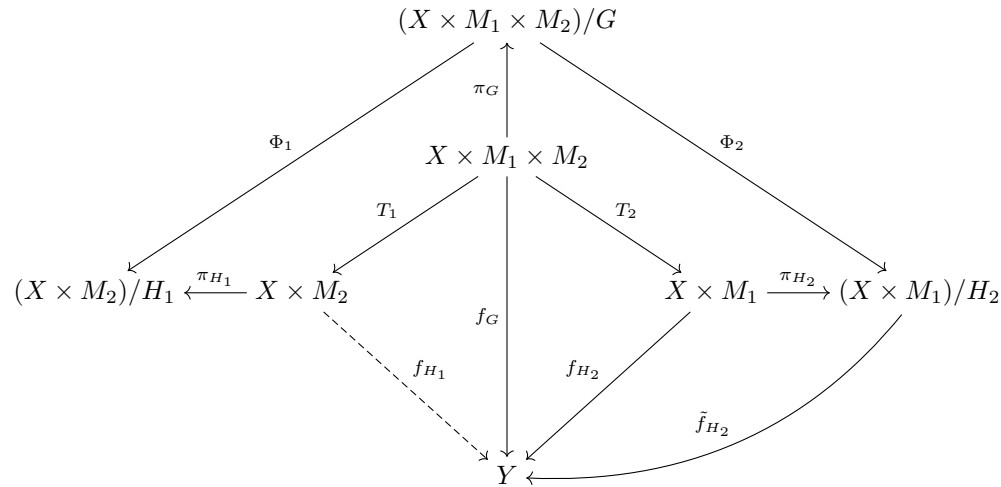

for $f_G : X \times M_1 \times M_2 \to Y$ the $G$ invariant given as in Equation (5), i.e., $f_G = f_{H_2} \circ T_2$.

*Proof.* First, we can see that $f_{H_1} = \tilde{f}_{H_2} \circ \Phi_2 \circ \Phi_1^{-1} \circ \pi_{H_1}$ is $H_1$ invariant since the projection map $\pi_{H_1}$ is $H_1$ invariant by definition.

Second, we will see that $f_G = f_{H_2} \circ T_2 = f_{H_1} \circ T_1$. Let $(p, q, x) \in M_1 \times X \times M_2$, then

$$
\begin{aligned}
f_{H_1} \circ T_1(x, p, q) &= f_{H_1}(\rho_1(p) \cdot x, \rho_1(p) \cdot q) \\
&= \tilde{f}_{H_2} \circ \Phi_2 \circ \Phi_1^{-1} \circ \pi_{H_1}(\rho_1(p) \cdot x, \rho_1(p) \cdot q) \\
&= \tilde{f}_{H_2} \circ \Phi_2 \circ \Phi_1^{-1} [H_1 \cdot (\rho_1(p) \cdot (x, q))] \\
&= \tilde{f}_{H_2} \circ \Phi_2 \circ \Phi_1^{-1} (G_{p,p_0} \cdot (x, q)) && \text{[by Equation (8)]} \\
&= \tilde{f}_{H_2} \circ \Phi_2(G \cdot (x, p, q)) && \text{[by Equation (9)]} \\
&= \tilde{f}_{H_2}(G_{q,q_0} \cdot (x, p)) \\
&= \tilde{f}_{H_2} [H_2 \cdot (\rho_2(q) \cdot (x, p))] \\
&= f_{H_2}(\rho_2(q) \cdot x, \rho_2(q) \cdot p) && \text{[by Theorem 2.1]} \\
&= f_{H_2} \circ T_2(x, p, q).
\end{aligned}
$$

$\square$

### C.2.2   PROOF FOR CHARACTERIZATION WHEN $M = G$

**Corollary 2.2 (Restated).** *Let the space $X \times G$, then the $G$ invariants $f_G : X \times G \to Y$ are characterized as*

$$
f_G(x, g) = f(g^{-1} \cdot x) \in Y, \quad \text{for some } f : X \to Y.
$$

*Proof.* First, notice that any group $G$ is a homogeneous space of itself, i.e., $G$ acts transitively on itself by left multiplication. Moreover, $G$ acts freely on itself, i.e., $H = \mathrm{Stab}_G(g) = \{e\}$ for any $g \in G$. Then, by Theorem 2.2, given the (unique) canonicalization map $\rho(g) = \rho(gH) = g^{-1}$, then the diagram

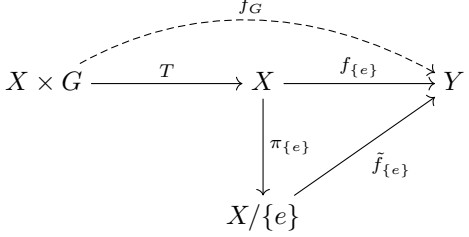

commutes for any arbitrary map $f_{\{e\}} : X \to Y$.

$\square$

### C.2.3   Iterated Reduction via Chains of Subgroups

In structured settings, the reduction can be applied iteratively, further simplifying invariant computations.

**Definition C.1** (Descending chain of homogeneous spaces)**.** *A descending chain of homogeneous spaces for a Lie group $G$ is a nested sequence of closed subgroups*

$$G = G_0 \supset G_1 \supset \cdots \supset G_k,$$

*yielding homogeneous spaces $M_i := G_i/G_{i+1}$ for $i = 0, \ldots, k-1$.*

**Lemma C.1** (Iterated reduction)**.** *Given a descending chain as in Definition C.1, there are bijections*

$$(X \times M_0 \times M_1 \times \cdots \times M_{k-1})/G_0 \longleftrightarrow (X \times M_1 \times \cdots \times M_{k-1})/G_1 \longleftrightarrow \cdots \longleftrightarrow X/G_k.$$

**Example C.1.** *Since $\mathbb{R}^3 \cong SE(3)/SO(3)$, $S^2 \cong SO(3)/SO(2)$, and $S^1 \cong SO(2)/\{e\}$, we have the descending chain*

$$SE(3) \supset SO(3) \supset SO(2) \supset \{e\}.$$

*Lemma C.1 yields the bijections*

$$(S^1 \times \mathbb{R}^3 \times S^2)/SE(3) \longleftrightarrow (S^1 \times S^2)/SO(3) \longleftrightarrow S^1/SO(2) \longleftrightarrow \{*\}.$$

*Consequently, every $SE(3)$-invariant on $S^1 \times \mathbb{R}^3 \times S^2$ is constant.*

## D   Computation of Invariants

In this section, we derive explicit separating invariants for Equivariant Neural Fields (ENFs) by instantiating the theoretical framework developed in Section 2.1. We focus on the Equivariant Neural Eikonal Solver introduced by García-Castellanos et al. (2025), extending its applicability beyond the original setting to arbitrary homogeneous conditioning spaces.

### D.1   Theoretical Setup

We begin by describing the full structure of the latent space in ENFs and clarifying that it is equivalent to the simplified framework presented in Section 3.

**Latent Space Decomposition.**   Following Wessels et al. (2024), an ENF represents a family of signals via a coordinate-based network $f_\theta : \mathcal{X} \times \mathcal{Z} \to \mathbb{R}^{d_{\text{out}}}$, where $\mathcal{X}$ denotes the input space (e.g., spatial coordinates) and $\mathcal{Z}$ is the *latent space* encoding the conditioning variable. In the main text, we use $\mathcal{Z}$ to denote this latent space for notational simplicity. However, in practice, $\mathcal{Z}$ admits a natural decomposition:

$$\mathcal{Z} := \mathcal{P} \times \mathcal{C}, \tag{10}$$

where:

- $\mathcal{P}$ is the *pose space*, carrying a nontrivial $G$-action. Elements $p \in \mathcal{P}$ are meant to encode geometric information (e.g., reference frames, positions, or orientations) that transform under the group.

- $\mathcal{C} = \mathbb{R}^d$ is the *context space*, consisting of context vectors $\mathbf{c} \in \mathbb{R}^d$ that are meant to encode non-geometric attributes (e.g., learned features). The group $G$ acts *trivially* on $\mathcal{C}$: for all $g \in G$ and $\mathbf{c} \in \mathcal{C}$, we have $g \cdot \mathbf{c} = \mathbf{c}$.

Under this decomposition, the diagonal $G$-action on $\mathcal{Z} = \mathcal{P} \times \mathcal{C}$ takes the form $g \cdot (p, \mathbf{c}) = (g \cdot p, \mathbf{c})$.

**Reduction to Pose-Space Invariants.** The trivial action on $\mathcal{C}$ has an important consequence: when constructing $G$-invariant functions on $\mathcal{X}^{\times m} \times \mathcal{Z}$, the appearance component $\mathbf{c}$ may be treated as a fixed parameter. Formally, a function $f_G : \mathcal{X}^{\times m} \times \mathcal{P} \times \mathcal{C} \to Y$ is $G$-invariant if and only if, for each fixed $\mathbf{c} \in \mathcal{C}$, the restricted function $f_G(\cdot, \cdot, \mathbf{c}) : \mathcal{X}^{\times m} \times \mathcal{P} \to Y$ is $G$-invariant. This allows us to identify the problem of computing $G$-invariants on $\mathcal{X}^{\times m} \times \mathcal{Z}$ with that of computing $G$-invariants on $\mathcal{X}^{\times m} \times \mathcal{P}$. Therefore, the appearance vector $\mathbf{c}$ can then be incorporated as an additional input to the network without affecting the symmetry structure.

**Application of Theorem 2.2.** With this simplification, we apply Generalized Reduction to the Isotropy. When the pose space is a homogeneous space $\mathcal{P} = G/H$ for some subgroup $H \leq G$, Theorem 2.2 establishes:

$$G\text{-invariants on } \mathcal{X}^{\times m} \times G/H \quad \longleftrightarrow \quad H\text{-invariants on } \mathcal{X}^{\times m}. \tag{11}$$

This correspondence is computationally significant: while $G$-invariants on the heterogeneous product $\mathcal{X}^{\times m} \times G/H$ may be difficult to characterize directly, $H$-invariants on $\mathcal{X}^{\times m}$ often fall within the scope of classical results from invariant theory. In particular, when $\mathcal{X} = \mathbb{R}^n$ and $H$ is a compact linear group, Weyl's First Fundamental Theorem provides a complete characterization of generating invariants.

### D.2    COMPUTATION STRATEGY

We now provide a detailed exposition of the procedure outlined in Algorithm 1, which we apply uniformly throughout this appendix.

1. **Specify the pose space.** Given a group $G$ acting on $\mathcal{X}$, choose the pose space $\mathcal{P} = G/H$ by selecting an appropriate subgroup $H \leq G$. Different choices yield latent representations with varying geometric content, e.g.:
   - $\mathcal{P} = G$ (i.e., $H = \{e\}$): full group as pose space.
   - $\mathcal{P} = \mathbb{R}^n \cong \mathrm{E}(n)/\mathrm{O}(n)$: position-only encoding.
   - $\mathcal{P} = \mathbb{R}^n \times S^{n-1} \cong \mathrm{E}(n)/\mathrm{O}(n-1)$: position-orientation encoding.
2. **Construct the canonicalization map.** Per Theorem 2.2, we require $\rho : \mathcal{P} \to G$ satisfying $\rho(p) \cdot p = p_0$ for a fixed reference point $p_0 \in \mathcal{P}$. When $\mathcal{P} = G/H$, the natural choice is $\rho(gH) = g^{-1}$, for any representative $g$ of $gH$.
3. **Compute $H$-invariants on $\mathcal{X}^{\times m}$.** By the correspondence (11), we reduce to computing $H$-invariants. For $\mathcal{X} = \mathbb{R}^n$, we appeal to Theorems D.1 and D.2. For embedded submanifolds $\mathcal{X} \subset \mathbb{R}^n$ for which the group action extends to an action on $\mathbb{R}^n$ (e.g., $S^{n-1}$ with $\mathrm{SO}(n)$), invariants are restrictions of the corresponding ambient-space invariants.
4. **Lift to $G$-invariants.** The $G$-invariants on $\mathcal{X}^{\times m} \times \mathcal{P}$ are obtained by precomposing the $H$-invariants with the map $T(x_1, \ldots, x_m, p) = (\rho(p) \cdot x_1, \ldots, \rho(p) \cdot x_m)$.

### D.3    FIRST FUNDAMENTAL THEOREMS

We recall two classical results that underpin our constructions.

**Theorem D.1** (First Fundamental Theorem of $\mathrm{O}(d)$; see (Weyl, 1946; Dym & Gortler, 2023)). *Let $n \geq d$. The set of separating invariants of $(\mathbb{R}^d)^{\times n}$ with respect to $\mathrm{O}(d)$ is given by*

$$\left\{ \|x_i\|_2^2 \right\}_{i=1,\ldots,n} \quad \bigcup \quad \left\{ \|x_i - x_j\|_2^2 \right\}_{1 \leq i < j \leq n}.$$

**Theorem D.2** (First Fundamental Theorem of $\mathrm{SO}(d)$; see (Weyl, 1946; Dym & Gortler, 2023)). *Let $n \geq d$. The set of separating invariants of $(\mathbb{R}^d)^{\times n}$ with respect to $\mathrm{SO}(d)$ is given by*

$$\left\{ \|x_i\|_2^2 \right\}_{i=1,\ldots,n} \quad \bigcup \quad \left\{ \|x_i - x_j\|_2^2 \right\}_{1 \leq i < j \leq n} \quad \bigcup \quad \{\det(x_{i_1}, x_{i_2}, \ldots, x_{i_d})\}_{1 \leq i_1 < i_2 < \cdots < i_d \leq n}.$$

**Remark D.1** (Generators vs. Functionally Independent Invariants). *Weyl's First Fundamental Theorems provide* generators *of the invariant ring, meaning every polynomial invariant can be expressed as a polynomial in these generators. However, generators need not be functionally independent: they may satisfy algebraic relations (syzygies). Two distinct notions are relevant:*

- **Orbit separation** *is required for* maximal expressivity*: by Proposition B.1, any continuous invariant function can be written as a composition of orbit-separating invariants with a continuous function. For compact groups, generators are orbit-separating (Dym & Gortler, 2023), so our constructions guarantee universal approximation when composed even with a simple MLP.*

- **Functional independence** *is required for* minimal representation*: the minimum number of functionally independent invariants equals the dimension of the orbit space, $m - s$, where $m = \dim(\mathcal{M})$ and $s = \dim(orbit)$ (Olver, 1995). This is relevant when minimizing the input dimension to downstream models.*

*Throughout this appendix, we prioritize orbit separation to ensure maximal expressivity. The invariant sets we provide are separating (and hence sufficient for universal approximation), though some may contain redundancies relative to the minimal functionally independent set.*

### D.4 EXPLICIT INVARIANTS

We now derive separating invariants for the Equivariant Neural Eikonal Solver (García-Castellanos et al., 2025) across various geometric configurations. In all cases, let $s, r \in \mathcal{X}$ denote source and receiver coordinates, and let $p \in \mathcal{P}$ denote the pose. Since $G$ acts isometrically on $\mathcal{X}$, the reduction to the isotropy translates the problem of computing separating invariants on a heterogeneous product space to computing invariants on spaces where a First Fundamental Theorem is available.

**2D Euclidean Input Space** Let $s, r \in \mathcal{X} = \mathbb{R}^2$. Then we will consider the following two isometry groups:

(a) **Euclidean group:** Let $G = E(2)$. Then we have three homogeneous spaces as candidates for our latent conditioning pose-space

  (i) **2D Euclidean Latent Space:** Let $p \in \mathcal{P} = \mathbb{R}^2 \equiv E(2)/O(2)$. By Weyl's First Main Theorem for the Euclidean group (Weyl, 1946; Olver, 2001), we have that a set of separating invariants of $\mathbb{R}^2 \times \mathbb{R}^2 \times \mathbb{R}^2$ with regards of $E(2)$ is

$$\boxed{\left\{ \|s - r\|_2^2, \|s - p\|_2^2, \|r - p\|_2^2 \right\}}$$

  **Remark D.2.** *In this case, we do not need to apply the reduction to the isotropy. However, as a verification step, we can confirm our result using reduction to the isotropy (Theorem 2.2), which establishes a one-to-one correspondence between $E(2)$-invariants on $\mathbb{R}^2 \times \mathbb{R}^2 \times E(2)/O(2)$ and $O(2)$-invariants on $\mathbb{R}^2 \times \mathbb{R}^2$.*
  *Indeed, when we take the set of $O(2)$-invariants on $\mathbb{R}^2 \times \mathbb{R}^2$ presented in Theorem D.1 and precompose it with the canonicalization obtained by subtracting $p$ from both $s$ and $r$, we recover precisely the same set of invariants given by Weyl's First Main Theorem for the Euclidean group.*

  (ii) **2D Position-Orientation Latent Space:** Let $p \in \mathcal{P} = \mathbb{R}^2 \times S^1 \equiv E(2)/O(1)$. Then by *reduction to the isotropy* (Theorem 2.2), $E(2)$-invariants on $\mathbb{R}^2 \times \mathbb{R}^2 \times E(2)/O(1)$ are in one-to-one correspondence with $O(1)$-invariants on $\mathbb{R}^2 \times \mathbb{R}^2$. Since $O(1)$ as the stabilizer subgroup of $((1,0),(1,0)) \in \mathbb{R}^2 \times S^1$ can be described as

$$O(1) \cong \left\{ \begin{pmatrix} 1 & 0 \\ 0 & 1 \end{pmatrix}, \begin{pmatrix} 1 & 0 \\ 0 & -1 \end{pmatrix} \right\}.$$

  Hence, the set of separating invariants of $\mathbb{R}^2 \times \mathbb{R}^2$ with regards $O(1)$ will be

$$\left\{ s_{(1)}, r_{(1)} \right\} \cup (\text{set of separating invariants of } \mathbb{R} \times \mathbb{R} \text{ with regards to } O(1)),$$

  where $s_{(1)}, r_{(1)}$ are the first coordinate of $s$ and $r$ respectively. By Theorem D.1 we know what are the set of separating invariants of $\mathbb{R} \times \mathbb{R}$ with regards $O(1)$. Meaning, that the set of separating invariants of $\mathbb{R}^2 \times \mathbb{R}^2$ with regards $O(1)$ will be

$$\boxed{\left\{ s_{(1)}, r_{(1)}, \left| s_{(2)} \right|^2, \left| r_{(2)} \right|^2, \left| s_{(2)} - r_{(2)} \right|^2 \right\} = \left\{ f_{O(1)}^\ell (s, r) \right\}_{\ell = 1, \ldots, 5}}$$

Let $p = (t, \alpha) \in \mathbb{R}^2 \times S^1$, as stated in Remark 2.1, we can identify it with a coset

$$p \leftrightarrow \bar{p}O(1) = \begin{pmatrix} & A & t_{(1)} \\ & & t_{(2)} \\ 0 & 0 & 1 \end{pmatrix} O(1) \in E(2)/O(1), \text{ with } A = \begin{pmatrix} \alpha_{(1)} & -\alpha_{(2)} \\ \alpha_{(2)} & \alpha_{(1)} \end{pmatrix} \in O(2).$$

Then we will construct the canonalization function $\rho : \mathbb{R}^2 \times S^1 \to E(2)$ satisfying the identity of Equation (4), as

$$\rho(p) = \bar{p}^{-1} = \begin{pmatrix} A^T & -A^T t \\ 0 & 0 & 1 \end{pmatrix} \in E(2).$$

Then, by Theorem 2.2, the set of separating invariants of $\mathbb{R}^2 \times \mathbb{R}^2 \times (\mathbb{R}^2 \times S^1)$ with regards to $E(2)$ is

$$\left\{ f_{O(1)}^\ell \left( A^T s - A^T t, A^T r - A^T t \right) \right\}_{\ell = 1, \ldots, 5}.$$

(iii) **Group Latent Space:** Let $p \in \mathcal{P} = E(2)$. By Corollary 2.2, the set of separating invariants of $\mathbb{R}^2 \times \mathbb{R}^2 \times E(2)$ is

$$\boxed{\left\{ p^{-1} \cdot s, p^{-1} \cdot r \right\}}$$

(b) **Special Euclidean group:** Let $G = SE(2)$. Then we have two homogeneous spaces as candidates for our latent conditioning pose-space

(i) **2D Euclidean Latent Space:** Let $p \in \mathcal{P} = \mathbb{R}^2 \equiv SE(2)/SO(2)$. By Weyl's First Main Theorem for the Special Euclidean group (Weyl, 1946; Olver, 2001), we have that a set of separating invariants of $\mathbb{R}^2 \times \mathbb{R}^2 \times \mathbb{R}^2$ with regards of $SE(2)$ is

$$\boxed{\left\{ \|s - r\|_2^2, \|s - p\|_2^2, \|r - p\|_2^2, \det(s - p, r - p) \right\}}$$

**Remark D.3.** *Again, in this case, we do not need to apply the reduction to the isotropy. However, as a verification step, we can confirm our result using reduction to the isotropy (Theorem 2.2), which establishes a one-to-one correspondence between $SE(2)$-invariants on $\mathbb{R}^2 \times \mathbb{R}^2 \times SE(2)/SO(2)$ and $SO(2)$-invariants on $\mathbb{R}^2 \times \mathbb{R}^2$. Indeed, when we take the set of $SO(2)$-invariants on $\mathbb{R}^2 \times \mathbb{R}^2$ presented in Theorem D.2 and precompose it with the canonicalization obtained by subtracting $p$ from both $s$ and $r$, we recover precisely the same set of invariants given by Weyl's First Main Theorem for the Special Euclidean group.*

(ii) **Group Latent Space:** Let $p \in \mathcal{P} = \mathbb{R}^2 \times S^1 \equiv SE(2)/SO(1) \equiv SE(2)/\{I\} \equiv SE(2)$. By Corollary 2.2, the set of separating invariants of $\mathbb{R}^2 \times \mathbb{R}^2 \times E(2)$ is

$$\boxed{\left\{ p^{-1} \cdot s, p^{-1} \cdot r \right\}}$$

**3D Euclidean Input Space** Let $s, r \in \mathcal{X} = \mathbb{R}^3$. Then we will consider the following two isometry groups:

(a) **Euclidean group:** Let $G = E(3)$. Then we have four homogeneous spaces as candidates for our latent conditioning pose-space

(i) **3D Euclidean Latent Space:** Let $p \in \mathcal{P} = \mathbb{R}^3 \cong E(3)/O(3)$. Since any three points in $\mathbb{R}^3$ lie in a common plane, a triangle in $\mathbb{R}^3$ is uniquely determined up to $E(3)$ by its three edge lengths. By Weyl's First Main Theorem for the Euclidean group (Weyl, 1946; Olver, 2001), a set of separating invariants of $\mathbb{R}^3 \times \mathbb{R}^3 \times \mathbb{R}^3$ with respect to $E(3)$ is

$$\boxed{\left\{ \|s - r\|_2^2, \|s - p\|_2^2, \|r - p\|_2^2 \right\}.}$$

(ii) **3D Position-Orientation Latent Space:** Let $p \in \mathcal{P} = \mathbb{R}^3 \times S^2 \equiv E(3)/O(2)$. Then by *reduction to the isotropy* (Theorem 2.2), $E(3)$-invariants on $\mathbb{R}^3 \times \mathbb{R}^3 \times E(3)/O(2)$ are in one-to-one correspondence with $O(2)$-invariants on $\mathbb{R}^3 \times \mathbb{R}^3$. Since $O(2)$ as the stabilizer subgroup of $((1,0,0),(1,0,0)) \in \mathbb{R}^3 \times S^2$ can be described as

$$O(2) \cong \left\{ \begin{pmatrix} 1 & 0 & 0 \\ 0 & & \\ 0 & & R \end{pmatrix} \: : \: R \in O(2) \right\} \quad \text{(it will fix } s_{(1)} \text{ and act on } (s_{(2)}, s_{(3)}))$$

Hence, the set of separating invariants of $\mathbb{R}^3 \times \mathbb{R}^3$ with regards $O(2)$ will be

$$\left\{ s_{(1)}, r_{(1)} \right\} \cup (\text{set of separating invariants of } \mathbb{R}^2 \times \mathbb{R}^2 \text{ with regards to } O(2)),$$

where $s_{(1)}, r_{(1)}$ are the first coordinate of $s$ and $r$ respectively. By Theorem D.1 we know what are the set of separating invariants of $\mathbb{R}^2 \times \mathbb{R}^2$ with regards $O(2)$. Meaning, that the set of separating invariants of $\mathbb{R}^3 \times \mathbb{R}^3$ with regards $O(2)$ will be

$$\boxed{\left\{ s_{(1)}, r_{(1)}, \left\| (s_{(2)}, s_{(3)}) \right\|_2^2, \left\| (r_{(2)}, r_{(3)}) \right\|_2^2, \left\| (s_{(2)} - r_{(2)}, s_{(3)} - r_{(3)}) \right\|_2^2 \right\} = \left\{ f_{O(2)}^\ell(s, r) \right\}_{\ell=1,\ldots,5}}$$

Let $p = (t, \alpha) \in \mathbb{R}^3 \times S^2$, as stated in Remark 2.1, we can identify it with a coset

$$p \leftrightarrow \bar{p}O(2) = \begin{pmatrix} & A & & t \\ & & & \\ 0 & 0 & 0 & 1 \end{pmatrix} O(2) \in E(3)/O(2),$$

with $A = \begin{bmatrix} | & | & | \\ \alpha & \beta & \alpha \times \beta \\ | & | & | \end{bmatrix} \in O(3)$, s.t. $\beta \perp \alpha$ and $\|\beta\|_2 = 1$.

Then we will construct the canonalization function $\rho : \mathbb{R}^3 \times S^2 \to E(3)$ satisfying the identity of Equation (4), as

$$\rho(p) = \bar{p}^{-1} = \begin{pmatrix} & A^T & & -A^T t \\ & & & \\ 0 & 0 & 0 & 1 \end{pmatrix} \in E(3). \tag{12}$$

Then, by Theorem 2.2, the set of separating invariants of $\mathbb{R}^3 \times \mathbb{R}^3 \times (\mathbb{R}^3 \times S^2)$ with regards to $E(3)$ is

$$\left\{ f_{O(2)}^\ell \left( A^T s - A^T t, A^T r - A^T t \right) \right\}_{\ell=1,\ldots,5}.$$

(iii) **Affine Stiefel Manifold Latent Space:** Let $p \in \text{Vaff}_{2,3} := \mathbb{R}^3 \times V_{2,3} = \{(t, F) \in \mathbb{R}^3 \times \mathbb{R}^{3 \times 2} \mid F^T F = I_2 \in \mathbb{R}^{2 \times 2}\}$, where $V_{2,3}$ is the Stiefel manifold of 2-frames in $\mathbb{R}^3$. As shown in Lim et al. (2021), this is a homogeneous space, i.e., $\text{Vaff}_{2,3} \equiv E(3)/O(1)$.

**Remark D.4.** *The affine Stiefel manifold $\text{Vaff}_{2,3} = \mathbb{R}^3 \times V_{2,3}$ provides a rich geometric interpretation of the learned latent pose $p$.*
*For any 2-frame $F = [\alpha \mid \beta] \in V_{2,3}$, the orthonormality conditions imply that $\alpha$ defines a point on $S^2$ while $\beta$, being perpendicular to $\alpha$, defines a unit tangent vector at that point. This identifies $V_{2,3}$ with the unit tangent bundle $UTS^2$.*
*Extending to $\text{Vaff}_{2,3}$, any element $(t, F)$ encodes:*

- *$t \in \mathbb{R}^3$: position*
- *$\alpha \in S^2$: orientation*
- *$\beta \in T_\alpha S^2$: instantaneous direction of orientation change*

*This structure naturally captures a hierarchy of motion information, not only where an object is and how it is oriented, but also the direction in which its orientation is evolving.*

By *reduction to the isotropy* (Theorem 2.2), $E(3)$-invariants on $\mathbb{R}^3 \times \mathbb{R}^3 \times E(3)/O(1)$ are in one-to-one correspondence with $O(1)$-invariants on $\mathbb{R}^3 \times \mathbb{R}^3$. Since $O(1)$ as the stabilizer subgroup of $((1,0,0),[(1,0,0)^T \mid (0,1,0)^T]) \in \mathrm{Vaff}_{2,3}$ can be described as

$$O(1) \cong \left\{ \begin{pmatrix} 1 & 0 & 0 \\ 0 & 1 & 0 \\ 0 & 0 & 1 \end{pmatrix}, \begin{pmatrix} 1 & 0 & 0 \\ 0 & 1 & 0 \\ 0 & 0 & -1 \end{pmatrix} \right\}.$$

Hence, the set of separating invariants of $\mathbb{R}^3 \times \mathbb{R}^3$ with regards $O(1)$ will be

$$\left\{ s_{(1)}, r_{(1)}, s_{(2)}, r_{(2)} \right\} \cup (\text{set of separating invariants of } \mathbb{R} \times \mathbb{R} \text{ with regards to } O(1))$$

Thus, by Theorem D.1, the set of separating invariants of $\mathbb{R}^3 \times \mathbb{R}^3$ with regards $O(1)$ will be

$$\boxed{\left\{ s_{(1)}, r_{(1)}, s_{(2)}, r_{(2)}, \left| s_{(3)} \right|^2, \left| r_{(3)} \right|^2, \left| s_{(3)} - r_{(3)} \right|^2 \right\} = \left\{ f_{O(1)}^{\ell}(s,r) \right\}_{\ell = 1, \dots, 7}}$$

Let $p = (t, F) \in \mathrm{Vaff}_{2,3}$ with $F = [\alpha \mid \beta] \in V_{2,3}$, as stated in Remark 2.1, we can identify it with a coset

$$p \leftrightarrow \bar{p}O(1) = \begin{pmatrix} & & & | \\ & A & & t \\ & & & | \\ 0 & 0 & 0 & 1 \end{pmatrix} O(1) \in E(3)/O(1), \text{ with } A = \begin{bmatrix} | & | & | \\ \alpha & \beta & \alpha \times \beta \\ | & | & | \end{bmatrix} \in O(3)$$

Then we will construct the canonalization function $\rho : \mathbb{R}^3 \times S^2 \to E(3)$ satisfying the identity of Equation (4), as $\rho(p) = \bar{p}^{-1}$

Then, by Theorem 2.2, the set of separating invariants of $\mathbb{R}^3 \times \mathbb{R}^3 \times (\mathbb{R}^3 \times S^2)$ with regards to $E(3)$ is

$$\left\{ f_{O(1)}^{\ell} \left( A^T s - A^T t, A^T r - A^T t \right) \right\}_{\ell = 1, \dots, 5}.$$

(iv) **Group Latent Space:** Let $p \in \mathcal{P} = E(3)$. By Corollary 2.2, the set of separating invariants of $\mathbb{R}^3 \times \mathbb{R}^3 \times E(3)$ is

$$\boxed{\left\{ p^{-1} \cdot s, p^{-1} \cdot r \right\}}$$

(b) **Special Euclidean group:** Let $G = SE(3)$. We consider three homogeneous spaces as candidates for our latent conditioning pose-space:

(i) **3D Euclidean Latent Space:** Let $p \in \mathcal{P} = \mathbb{R}^3 \equiv SE(3)/SO(3)$. In this case, $E(3)$-invariants of $\mathbb{R}^3 \times \mathbb{R}^3 \times \mathbb{R}^3$ are in one-to-one correspondence with their $SE(3)$-invariants. This is formalized as follows:

**Lemma D.1.** *Let $t \in (\mathbb{R}^3)^{\times 3}$ be a triplet of 3D points. The orbit of $t$ under $SE(3)$ is the same as its orbit under $E(3)$.*

*Proof.* **Step 1: Coplanarity and choice of reflection.** Any three points in $\mathbb{R}^3$ lie in a unique affine plane $\Pi$. Let

$$f : \mathbb{R}^3 \longrightarrow \mathbb{R}^3$$

be the reflection in the plane $\Pi$. By construction:

$$f^2 = \mathrm{id}, \quad \det(\text{Jacobian matrix of} f) = -1, \quad f(t_i) = t_i \quad (\forall\, i = 1, 2, 3).$$

Hence $f \in E(3) \setminus SE(3)$ and $f\, t = t$.

**Step 2: Index-2 subgroup and coset decomposition.** The determinant map

$$\det : E(3) \longrightarrow \{\pm 1\}, \qquad (x \mapsto Rx + b) \mapsto \det(R)$$

is a surjective homomorphism whose kernel is exactly $SE(3)$. Therefore

$$[E(3) : SE(3)] = 2, \quad SE(3) \lhd E(3).$$

Choosing the reflection $f \notin SE(3)$ as above, we obtain the set-product decomposition

$$E(3) = SE(3)\{e, f\} = \{e, f\} SE(3).$$

**Step 3: Orbits coincide.** Acting on $t$, we have

$$E(3)\, t = \big(SE(3)\{e, f\}\big)\, t = SE(3)\big(\{e, f\}\, t\big).$$

Since $f\, t = t$, the set $\{e, f\}\, t = \{t\}$. Hence

$$E(3)\, t = SE(3)\{t\} = SE(3)\, t.$$

$\square$

**Corollary D.1.** *Let $T = (\mathbb{R}^3)^{\times 3}$ be the space of triplets of 3D points. The orbit spaces $T/SE(3)$ and $T/E(3)$ are identical.*

**Corollary D.2.** *Any $SE(3)$-invariant function $f_{SE(3)} : T \to \mathbb{R}$ is also $E(3)$-invariant.*

By Corollary D.2, $SE(3)$-orbits and $E(3)$-orbits coincide for three points in $\mathbb{R}^3$. Therefore, the set of separating invariants of $\mathbb{R}^3 \times \mathbb{R}^3 \times \mathbb{R}^3$ with respect to $SE(3)$ is

$$\boxed{\left\{ \|s - r\|_2^2, \|s - p\|_2^2, \|r - p\|_2^2 \right\}}$$

This result can be generalized using the following lemma:

**Lemma D.2.** *Let $G$ be a group acting on a set $X$. Let $x \in X$ and define $H = \operatorname{Stab}_G(x)$. Suppose we have the decomposition $G = G'H$ for some subgroup $G'$. Then $Gx = G'x$.*

*Proof.* $Gx = G'Hx = G'x$. $\square$

**Corollary D.3.** *Let $K_n = (\mathbb{R}^n)^{\times n}$ be the space of $n$-tuples of $\mathbb{R}^n$ points. The orbit spaces $K_n/SE(n)$ and $K_n/E(n)$ are identical. Therefore, any $SE(n)$-invariant function $f_{SE(n)} : K_n \to \mathbb{R}$ is also $E(n)$-invariant.*

(ii) **3D Position-Orientation Latent Space:** Let $p \in \mathcal{P} = \mathbb{R}^3 \times S^2 \equiv SE(3)/SO(2)$. Then by *reduction to the isotropy* (Theorem 2.2), $SE(3)$-invariants on $\mathbb{R}^3 \times \mathbb{R}^3 \times SE(3)/SO(2)$ are in one-to-one correspondence with $SO(2)$-invariants on $\mathbb{R}^3 \times \mathbb{R}^3$. Since $SO(2)$ as the stabilizer subgroup of $((1,0,0),(1,0,0)) \in \mathbb{R}^3 \times S^2$ can be described as

$$SO(2) \cong \left\{ \begin{pmatrix} 1 & 0 & 0 \\ 0 & & \\ 0 & & R \end{pmatrix} : R \in SO(2) \right\} \quad \left( \text{it will fix } s_{(1)} \text{ and act on } (s_{(2)}, s_{(3)}) \right)$$

Hence, the set of separating invariants of $\mathbb{R}^3 \times \mathbb{R}^3$ with regards $SO(2)$ will be

$$\left\{ s_{(1)}, r_{(1)} \right\} \cup (\text{set of separating invariants of } \mathbb{R}^2 \times \mathbb{R}^2 \text{ with regards to } SO(2)),$$

where $s_{(1)}, r_{(1)}$ are the first coordinate of $s$ and $r$ respectively. By Theorem D.2 we know what are the set of separating invariants of $\mathbb{R}^2 \times \mathbb{R}^2$ with regards $SO(2)$. Meaning, that the set of separating invariants of $\mathbb{R}^3 \times \mathbb{R}^3$ with regards $SO(2)$ will be

$$\boxed{\begin{aligned} &\left\{ s_{(1)}, r_{(1)}, \left\| (s_{(2)}, s_{(3)}) \right\|_2^2, \left\| (r_{(2)}, r_{(3)}) \right\|_2^2, \right. \\ &\left. \left\| (s_{(2)} - r_{(2)}, s_{(3)} - r_{(3)}) \right\|_2^2, \det((s_{(2)}, s_{(3)}), (r_{(2)}, r_{(3)})) \right\} = \left\{ f_{SO(2)}^\ell(s, r) \right\}_{\ell = 1, \ldots, 6} \end{aligned}}$$

Let $p = (t, \alpha) \in \mathbb{R}^3 \times S^2$, as stated in Remark 2.1, we can identify it with a coset

$$p \leftrightarrow \bar{p}SO(2) = \begin{pmatrix} & & & | \\ & A & & t \\ & & & | \\ 0 & 0 & 0 & 1 \end{pmatrix} SO(2) \in SE(3)/SO(2),$$

with $A = \begin{bmatrix} | & | & | \\ \alpha & \beta & \alpha \times \beta \\ | & | & | \end{bmatrix} \in SO(3)$, s.t. $\beta \perp \alpha$ and $\|\beta\|_2 = 1$.

One can check that with this construction $\|\alpha \times \beta\|_2 = 1$, and $\det(A) = 1$, i.e., $A \in SO(3)$. Then we will construct the canonalization function $\rho : \mathbb{R}^3 \times S^2 \to SE(3)$ satisfying the identity of Equation (4), as $\rho(p) = \bar{p}^{-1}$.

Thus, by Theorem 2.2, the set of separating invariants of $\mathbb{R}^3 \times \mathbb{R}^3 \times (\mathbb{R}^3 \times S^2)$ with regards to $E(3)$ is

$$\left\{ f_{SO(2)}^\ell \left( A^T s - A^T t, A^T r - A^T t \right) \right\}_{\ell = 1, \dots, 6}.$$

(iii) **Group Latent Space:** Let $p \in \mathcal{P} = SE(3)$. By Corollary 2.2, the set of separating invariants of $\mathbb{R}^3 \times \mathbb{R}^3 \times SE(3)$ is

$$\boxed{\left\{ p^{-1} \cdot s, p^{-1} \cdot r \right\}}$$

**Spherical Input Space** Let $s, r \in \mathcal{X} = S^2 \subset \mathbb{R}^3$. Then we will consider the following two isometry groups:

(a) **Orthogonal group:** Let $G = O(3)$. Then we have three homogeneous spaces as candidates for our latent conditioning pose-space

  (i) **Spherical Latent Space:** Let $p \in \mathcal{P} = S^2 \equiv O(3)/O(2)$. Since $S^2 \times S^2 \times S^2$ is an embedded sub-manifold of $\mathbb{R}^3 \times \mathbb{R}^3 \times \mathbb{R}^3$, we will have that the set of $O(3)$ separating invariants of $S^2 \times S^2 \times S^2$ will be the restrictions of those of $\mathbb{R}^3 \times \mathbb{R}^3 \times \mathbb{R}^3$ given by Theorem D.1, i.e.,

  $$\boxed{\left\{ \|s - r\|_2^2, \|s - p\|_2^2, \|r - p\|_2^2 \right\}} \quad \text{(the terms } \|s\|_2^2 = \|r\|_2^2 = \|p\|_2^2 = 1 \text{ are constant)}$$

  (ii) **Stiefel Manifold Latent Space:** Let $p \in V_{2,3} = \{F \in \mathbb{R}^{3 \times 2} \mid F^T F = I_2 \in \mathbb{R}^{2 \times 2}\} \equiv O(3)/O(1)$ (Lim et al., 2021). Then by *reduction to the isotropy* (Theorem 2.2), $O(3)$-invariants on $S^2 \times S^2 \times O(3)/O(1)$ are in one-to-one correspondence with $O(1)$-invariants on $S^2 \times S^2$. Since $O(1)$ as the stabilizer subgroup of $[(1,0,0)^T \mid (0,1,0)^T] \in V_{2,3}$ can be described as

  $$O(1) \cong \left\{ \begin{pmatrix} 1 & 0 & 0 \\ 0 & 1 & 0 \\ 0 & 0 & 1 \end{pmatrix}, \begin{pmatrix} 1 & 0 & 0 \\ 0 & 1 & 0 \\ 0 & 0 & -1 \end{pmatrix} \right\}.$$

  Hence, the set of separating invariants of $S^2 \times S^2$ with regard to $O(1)$ is obtained by restricting the $O(1)$ invariants of $\mathbb{R}^3 \times \mathbb{R}^3$ to the sphere. A minimal separating set is:

  $$\boxed{\left\{ s_{(1)}, s_{(2)}, r_{(1)}, r_{(2)}, s_{(3)} r_{(3)} \right\} = \left\{ f_{O(1)}^\ell (s, r) \right\}_{\ell = 1, \dots, 5}}$$

**Remark D.5.** *On $S^2$, the constraint $\|s\|_2^2 = 1$ implies $(s_{(3)})^2 = 1 - (s_{(1)})^2 - (s_{(2)})^2$, so $(s_{(3)})^2$ is determined by $s_{(1)}$ and $s_{(2)}$ (similarly for $r$). Moreover, $(s_{(3)} - r_{(3)})^2 = (s_{(3)})^2 - 2 s_{(3)} r_{(3)} + (r_{(3)})^2$, so the only independent information is $s_{(3)} r_{(3)}$. Thus a minimal separating set is $\{s_{(1)}, s_{(2)}, r_{(1)}, r_{(2)}, s_{(3)} r_{(3)}\}$.*

Let $p = [\alpha \mid \beta] \in V_{2,3}$, as stated in Remark 2.1, we can identify it with a coset

$$p \leftrightarrow \bar{p}O(1) = \begin{bmatrix} | & | & | \\ \alpha & \beta & \alpha \times \beta \\ | & | & | \end{bmatrix} O(1) \in O(3)/O(1),$$

Then we will construct the canonalization function $\rho : V_{2,3} \to O(3)$ satisfying the identity of Equation (4), as $\rho(p) = \bar{p}^{-1} = \bar{p}^T$.

Thus, by Theorem 2.2, the set of separating invariants of $S^2 \times S^2 \times V_{2,3}$ with regards to $O(3)$ is

$$\left\{ f^\ell_{O(1)} \left( \bar{p}^T \cdot s, \bar{p}^T \cdot r \right) \right\}_{\ell=1,\dots,5}.$$

(iii) **Group Latent Space:** Let $p \in \mathcal{P} = O(3)$. By Corollary 2.2, the set of separating invariants of $\mathbb{R}^2 \times \mathbb{R}^2 \times O(3)$ is

$$\boxed{\left\{ p^T \cdot s, p^T \cdot r \right\}}$$

(b) **Special Orthogonal group:** Let $G = SO(3)$. Then we have two homogeneous spaces as candidates for our latent conditioning pose-space

(i) **Spherical Latent Space:** Let $p \in \mathcal{P} = S^2 \equiv SO(3)/SO(2)$. The set of separating invariants (inhered from $\mathbb{R}^3 \times \mathbb{R}^3 \times \mathbb{R}^3$) are the restriction of the invariants given by Theorem D.2, i.e.,

$$\boxed{\left\{ \|s - r\|_2^2, \|s - p\|_2^2, \|r - p\|_2^2, \det(s, r, p) \right\}}$$

(ii) **Group Latent Space:** Let $p \in \mathcal{P} = SO(3)$. By Corollary 2.2, the set of separating invariants of $\mathbb{R}^2 \times \mathbb{R}^2 \times SO(3)$ is

$$\boxed{\left\{ p^T \cdot s, p^T \cdot r \right\}}$$

# E   FURTHER DISCUSSION

Symmetry-preserving neural network architectures have proven highly effective across diverse application domains, yet the two dominant paradigms, namely group convolutions and steerable networks, each exhibit fundamental limitations.

Group convolutional networks (Cohen & Welling, 2016) extend the translation equivariance of classical CNNs to richer symmetry groups but face fundamental constraints when the symmetry group contains continuous non-translational components. For example, achieving $SE(n)$-equivariance requires discretizing $SO(n)$ to a finite subgroup, an approximation that introduces errors and breaks exact equivariance guarantees (Weiler et al., 2018).

Steerable networks (Cohen et al., 2019) take a complementary approach rooted in representation theory: they decompose feature spaces into irreducible representations and parameterize equivariant linear maps via Schur's lemma. However, this formalism introduces two critical bottlenecks. First, preserving equivariance throughout the network requires specialized activation functions that respect the irreducible representation structure, constraining architectural flexibility and potentially limiting expressivity (Bekkers et al., 2023). Second, the explicit computation of Clebsch–Gordan coefficients, necessary to decompose tensor products into direct sums of irreducible representations, becomes computationally intractable for general groups[1] beyond well-studied cases such as $SO(3)$ (Villar et al., 2021).

Invariant-theoretic approaches offer a principled alternative that circumvents both difficulties. Methods such as those in Villar et al. (2021), Bekkers et al. (2023), and Blum-Smith et al. (2024) parameterize equivariant maps directly through a set of separating invariants rather than through irreducible decompositions. Crucially, this formulation achieves provable universality when composed

---

[1]Clebsch–Gordan coefficients can be computed for other Lie groups beyond $SO(3)$ (Alex et al., 2011), but such extensions remain an active area of research in tensor networks.

with any universal function approximator (see Proposition B.1), imposing virtually no architectural constraints and enabling practitioners to leverage state-of-the-art backbones (Blum-Smith & Villar, 2023). A caveat of such methods is that the number of invariants required for maximum expressivity grows with the number of input coordinates and the group's dimensionality, increasing computational cost. Nevertheless, sketching approaches such as Dym & Gortler (2023) demonstrate that random projections of set invariants can mitigate this expense.

Our proposed *Generalized Reduction to the Isotropy* (Theorem 2.2) extends the applicability of these invariant-theoretic methods to heterogeneous product spaces, enabling practitioners to leverage classical tools on the reduced space and lift the resulting invariants via canonicalization. We emphasize that with these separating invariants, one can construct *any* invariant function on heterogeneous products in a provably universal manner. However, extending this framework to fully equivariant architectures, particularly achieving universal approximation with equivariant aggregation mechanisms, remains an open problem; our method provides essential building blocks but does not yet offer a complete solution for the equivariant case.

It is worth noting that equivariant architectures themselves are a popular choice for learning invariant maps, since invariant functions form a subset of equivariant ones. In group convolutional networks, invariance is achieved via group pooling over the output, while in steerable networks, one simply restricts the output to type-0 (scalar) steerable features.

A natural application domain beyond Equivariant Neural Fields is *equivariant reinforcement learning*. When a Markov Decision Process (MDP) admits a symmetry group $G$, encoding the symmetry as an inductive bias in value functions can substantially improve sample efficiency and generalization (Wang et al., 2022). In a $G$-invariant MDP, the optimal action-value function satisfies

$$Q^*(s, a) = Q^*(g \cdot s, g \cdot a), \quad \forall g \in G,$$

motivating the construction of invariant Deep-Q networks. Our reduction framework is particularly well-suited to this setting for two reasons. First, practical RL state spaces are rarely homogeneous: they typically comprise spatial coordinates, orientations, internal configuration variables, and high-dimensional observations (e.g., images), each transforming differently under $G$ (Kober et al., 2013). Second, the $Q$-function is defined on the product space $\mathcal{S} \times \mathcal{A}$, which is itself a heterogeneous product, as states and actions often belong to qualitatively different representation spaces of $G$. Our framework provides a principled approach for constructing invariant value functions on such compound spaces. A systematic investigation of how different choices of invariant representations affect learning dynamics and generalization in this context constitutes an exciting direction for future work.

