# OpenReview forum: "Generalized Reduction to the Isotropy for Flexible Equivariant Neural Fields"
_ICLR.cc/2026/Workshop/GRaM — ICLR 2026 Workshop GRaM Poster_

### Official Review · Reviewer_KHBL · 2026-02-23
**Review of Generalized Reduction to the Isotropy for Flexible Equivariant Neural Fields**

**Rating:** 8
**Confidence:** 4

**Review:**

# Summary
This paper explores the construction of invariant functions on product spaces of form $X\times M$ where one of the spaces $M$ is transitive under group action $G$. Crucially, any invariant function can be viewed as a function on the orbits. By establishing an orbit equivalence $(X\times M)/G\cong X/H$, they reduce the original problem to constructing $H$ invariant functions on $X$. This equivalence effectively comes from canonicalization on $M$ with $H$ being the “leftover ambiguity” in transformations giving us orbits on $X$.

# Strengths and weaknesses
## Strengths
* The problem and results are presented in a clear and rigorous way.
* The appendix is extensive, providing clear proofs of the main results and additional generalizations and applications.
* The results are interesting, relatively intuitive, and seem generalizable to the fully equivariant case.

## Weaknesses
* Some notation is not explicitly defined (mostly Corollary 2.1 using \tilde to denote the function on orbits and $\pi_{H_1}$ to denote quotient map).
* There are no experiments (even if very small scale) of a model constructed using the methods described.

# Questions
* These results seem very generalizable to the fully equivariant case since the value of a function on an orbit representative fully characterizes the function on the entire orbit. Is this something you have considered?

**Pmlr Suitability:**

NA

---

### Official Review · Reviewer_tTkB · 2026-02-24
**Easy to read, but leaves wanting more: enough for tiny track.**

**Rating:** 7
**Confidence:** 3

**Review:**

The paper demonstrates a bijection between the orbit spaces (X \times M) / G and X / H for H isotropy subgroup, whenever G acts transitively on M. This allows one to construct G-invariant functions on the product space X \times M by reducing to H-invariants on X alone.

Overall, the paper is easy to read and understand and has extensive appendices, but while theoretical justifications are of significant interest, the practical significance of this lies primarily in being able to fundamentally justify canonicalization as a complete class of reductions for equivariance in the case of homogeneous spaces. I would encourage authors to focus more on the connection to canonicalization (move some of appendix A to main body), as this is precisely the class of methods reducing the problem of invariance to that of maps on orbits.

This further emphasizes that the primary issue with applying this method in practice is hidden away in the details in this paper: choice of p_0 is likely to have an enormous effect in practice.


Further Questions/Comments:
* I personally find the term "stabilizer subgroup" clearer than "isotropy subgroup", especially given that the main result of this paper is effectively an application of the orbit-stabilizer theorem.
* Page 2 line 102 you state that intuition for the result is in App C2: I would really suggest not dropping statements relating to intuition from main body, as they are precisely the ones that allow readers to understand the paper.
* I would really suggest relating to the canonicalization literature  (you seem to already try to relate to the moving frames in page 3 line 118, but ideally you could also relate to the recent notions you mention in App A.)
* Theorem 2.2: You state that "every G-invariant on X\timesM arises unqiuely in this manner: it would be useful to state precisely in what sense this is unique: its only unique for a fixed choice of p_0, right? This is precisely the part that is important in practice.
* Cor 2.1 What do you refer here to as T_1 and T_2?

**Pmlr Suitability:**

NA

---

### Meta-Review · Area_Chair_6v4Z · 2026-02-27

**Decision:**

Accept

**Metareview:**

I concur with the reviewer comments, and think this is a good paper for the tiny paper track. The authors are encouraged to incorporate reviewer comments in the revision.

**Relevance To Proceedings:**

Tiny paper — does not apply

**Relevance To Workshop:**

Yes — suitable for GRaM

---

### Decision · Program_Chairs · 2026-03-02

Accept (Poster)